# Effect of ripening time on the content of bioactive peptides and fatty acids profile of Artisanal *Coalho* cheese

**Débora A. F. V. A. Bezerra**[1], **Karoline M. S. Souza**[2], **Danielle C. Sales**[1]*, **Emmanuella O. M. Araújo**[1], **Stela A. Urbano**[1], **Claudio Cipolat-Gotet**[3], **Katya Anaya**[4], **Cláudio V. D. M. Ribeiro**[5], **Ana Lúcia F. Porto**[6], **Adriano H. N. Rangel**[1]

1 Academic Unit Specialized in Agricultural, Federal University of Rio Grande do Norte (UFRN), Macaiba, Rio Grande do Norte, Brazil, 2 Biosciences Center, Federal University of Pernambuco (UFPE), Recife, Pernambuco, Brazil, 3 Department of Veterinary Science, University of Parma, Parma, Italy, 4 Faculty of Health Sciences of Tmiri, Federal University of Rio Grande do Norte (UFRN), Santa Cruz, Rio Grande do Norte, Brazil, 5 School of Veterinary Medicine and Animal Science, Federal University of Bahia (UFBA), Salvador, Bahia, Brazil, 6 Morfology and Animal Fisiology Departament, Rural Federal University of Pernambuco (UFRPE), Recife, Pernambuco, Brazil

* daniellecsales@hotmail.com

**Data Availability Statement:** All relevant data are within the manuscript and its Supporting Information files.

## Abstract

The present study aimed to investigate the influence of ripening on the physicochemical, microbiological aspects, and fatty acid profile of Artisanal *Coalho* Cheeses and to detect if there are peptides with bioactive potential in their composition. Artisanal *Coalho* Cheese samples were kindly provided by a dairy farm located in Brazil in the Rio Grande do Norte state. A completely randomized design was adopted, with four maturation periods (0, 30, 45, and 60 days). Physicochemical traits (pH, total solids, moisture, non-fat solids, fat in total solids, protein, ash, fatty acid profile) and microbiological characterization (*Salmonella* sp, *Listeria monocytogenes*, total and thermotolerant coliforms, *Staphylococcus aureus*) were analyzed on cheese samples. Additionally, assays were performed for antioxidant and antihypertensive bioactivity through ACE and antimicrobial inhibition of the peptides extracted from the samples. There was a linear increase in total solids and ash content and a decrease in moisture content with increasing maturation time. The matured cheese samples had a lower pH than fresh Artisanal *Coalho* Cheese. Twenty-seven fatty acids were identified in the cheeses: 15 saturated, 07 monounsaturated, and 05 polyunsaturated, with a linear reduction of essential fatty acids (n6 and n3) during maturation. The microbiological quality of the cheeses was satisfactory, with an absence of undesirable bacteria in 92% of the cheese samples. Water-soluble peptide fractions from all periods tested showed antioxidant and antihypertensive potential with ACE control, and the maturation process potentiated these capacities, with a decline in these activities observed at 60 days. The antimicrobial activity against Gram-positive and Gram-negative bacteria increased with maturation, reaching better results until 60 days. The maturation process on wooden planks in the periods of 30, 45, and 60 days allows the production of Artisanal *Coalho* Cheese of an innovative character, safe to consumers from the microbiological point of view, with differentiated physicochemical and functional characteristics and good quality of lipid fraction compared to fresh cheese, enabling the addition of value to the dairy chain.

**Funding:** The author(s) received no specific funding for this work.

## Introduction

Several research studies have been carried out worldwide to search for foods characterized by compounds capable of improving and maintaining well-being and preventing chronic degenerative pathologies in humans [1].

Bioactive peptides constitute a class of protein molecules released/activated by enzymatic, microbial, and intestinal digestive hydrolysis and have beneficial pharmacological properties [2, 3]. The physiological functions regulated by bioactive peptides vary according to the amino acid sequence and their location within the protein [4]. They may exhibit antihypertensive, antimicrobial, anticarcinogenic, antioxidant, immunomodulatory, anticarcinogenic, and opioid properties, all of which have been identified in cheeses [4–9].

Lipids are important nutritional components found in most biological systems, animal, plant, or microbial, playing an important role in the composition and characteristics of cell membranes, providing energy, carrying fat-soluble vitamins (A, D, E, and K), and supplying the demand for essential fatty acids. Fatty acids originate from the breakdown of triglycerides. Those lipid molecules containing saturated and trans fatty acids are thus associated with various diseases and health conditions such as obesity, hypertension, cardiovascular disease, and cancer [10, 11]. Conversely, unsaturated fatty acids may lower blood cholesterol levels and promote weight control and hormonal regulation [12, 13].

The maturation of cheese encompasses several biochemical pathways that involve proteolytic, lipolytic, and glycolytic reactions. Many dairy cultures exhibit highly proteolytic activity, accumulating bioactive peptides in matured dairy products. The level of peptides naturally formed in the matrix depends on the type of dairy product and varies along with the balance between release and further hydrolysis during maturation [14].

Several authors have already pointed out that the maturation of cheeses can potentiate the formation and release of bioactive peptides, acting together with the enzymes produced by the initial and non-initial lactic acid bacteria [15–19]. During maturation, milk fat is also broken down, resulting in the release of fatty acids [20].

Artisanal *Coalho* Cheese stands out as one of the main artisanal cheeses produced in Brazil, with nutritional, social, cultural, and economic importance [21–23]. This cheese has characteristics that are well-defined by state and national standards. Despite this, there is also room for innovation in the market, such as adapting its process to producing fine and innovative cheeses (authorial and exclusive). These innovative cheeses can be matured, seasoned, and smoked with diverse characteristics, imprinting high quality and added value. The production of fine cheeses has increased significantly in recent years despite being offered at a higher price [24].

Matured Artisanal *Coalho* Cheese is an innovative product with high added value, with characteristics and sensory attributes distinct from fresh Artisanal *Coalho* Cheese; it has great potential to integrate and strengthen production in agro-industrial chains [25]. Thus, the present work aimed at investigating the influence of maturation on the physicochemical characteristics, microbiological aspects, fatty acid profile, and the presence of bioactive peptides in Artisanal *Coalho* Cheese.

## Materials and methods

### Raw milk

The raw milk used in the processing of Artisanal *Coalho* Cheeses production was supplied by a commercial farm in the São Gonçalo do Amarante, Rio Grande do Norte State, Brazil. The farm follows the principles of animal health, hygiene, and welfare when obtaining the raw

**Table 1. Physicochemical, microbiological characteristics and somatic cell count of the raw milk used to obtain Artisanal *Coalho* Cheese.**

| Characteristic | Raw milk[a] | Regulatory standard[b] |
|---|---|---|
| Fat (g 100/g) | 3.49 | Min. 3.0 |
| Total protein (g 100/g) | 2.98 | Min. 2.9 |
| Lactose (g 100/g) | 4.47 | Min. 4.3 |
| SNF (g 100/g) | 8.17 | Min. 8.4 |
| TS (g 100/g) | 11.63 | Min. 11.4 |
| Acidity (g of lactic acid/100 mL) | 0.17 | 0.14–0.18 |
| SCC (SC 100/mL) | 350,000 | Max. 400,000 |
| SPC (CFU 100/mL) | 700 | Max. 10,000 |

SNF: Solids-not-fat; TS: Total solids; SCC: Somatic Cell Count; SPC: Standard Plate Count (expressed as the number of colony forming units per milliliter). Analysis methods: Composition was determined by Infrared spectrophotometer (Dairy Spect®, Bentley Instruments Inc., Chaska, MN, USA); Acidity was determined by titration with Dornic solution according to AOAC [27]; SCC was determined by flow cytometry (BactoCounter IBCm®, Bentley Instruments Inc., Chaska, MN, USA).

[a]Mean of three repetitions.

[b]limits recommended by Normative Instruction No. 76 of November 26, 2018 [26].

milk. The fresh milk had physicochemical traits, somatic cell count (SCC), and standard plate count (SPC) within the standard established by Normative Instruction No. 76 of November 26, 2018 [26] (Table 1).

## Cheesemaking

The samples of Artisanal *Coalho* Cheeses were kindly provided by an artisanal cheese-producing farm in Brazil, Rio Grande do Norte, Municipality of São Gonçalo do Amarante.

The Artisanal *Coalho* Cheeses were made using raw milk up to 2 hours after milking, at a temperature of 35°C, following the traditional method used in this region [28]. Fig 1 shows the flowchart of the cheese manufacturing process in the experiment.

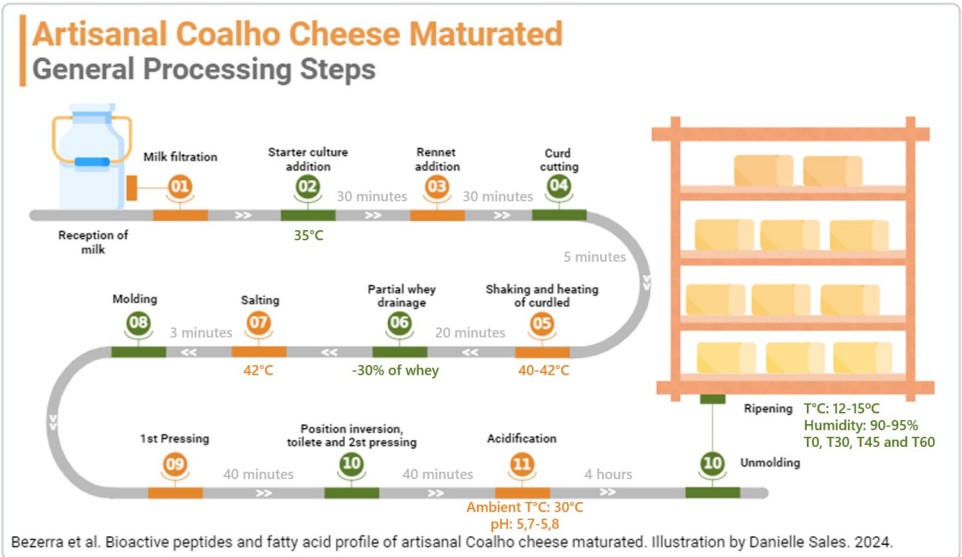

**Fig 1. Flowchart of the Artisanal *Coalho* Cheese manufacturing process in the experiment.**

The cheeses remained in the molds for 5 hours at room temperature to finish the acidification of the dough (pH around 5.7 to 5.8). Then, they were unmolded and distributed on wooden shelves in a maturation chamber, organized into groups according to the maturation times evaluated in the experiment. The conditions of the maturation chamber were controlled at 90 to 95% relative humidity at 12 to 15°C.

The cheeses were turned every three days in the first week and every seven days until the end of maturation. Every five days, the cheeses were washed with running water with the help of sponges to remove mold from their rind and, later, sprayed with 70% alcohol over the entire crust.

## Experimental groups

Artisanal *Coalho* Cheeses (n = 16) were produced from a batch of milk in 500g molds and divided into 4 maturation periods: 0 (fresh), 30, 45, and 60 days (Fig 2).

The physicochemical and microbiological characteristics of the cheeses of each period were evaluated. About 180 g of each cheese was removed, wherein 60 g was frozen to assess the fatty acid profile at the Multifunctional Laboratory of the School of Veterinary Medicine and Animal Science of the Federal University of Bahia (UFBA, Salvador/BA, Brazil) and the remaining 120 g were frozen for evaluation of the presence of bioactive peptides at the Laboratory of Bioactive Technology–LABTECBI of the Federal University of Pernambuco (UFPE, Recife/PE, Brazil).

## Physicochemical analysis

The analysis of pH, total solids, moisture, fat in total solids, protein, and ash of the cheeses was carried out in triplicate at the Animal Nutrition Laboratory of the Federal University of Rio Grande do Norte (UFRN, Macaíba/RN, Brazil).

## Hydrogen potential–pH

The hydrogen potential (pH) values were measured in cheese samples using a pH meter (Cap-Lab® model 210B). A 20 g portion of the crushed cheese sample was weighed with 20 mL of

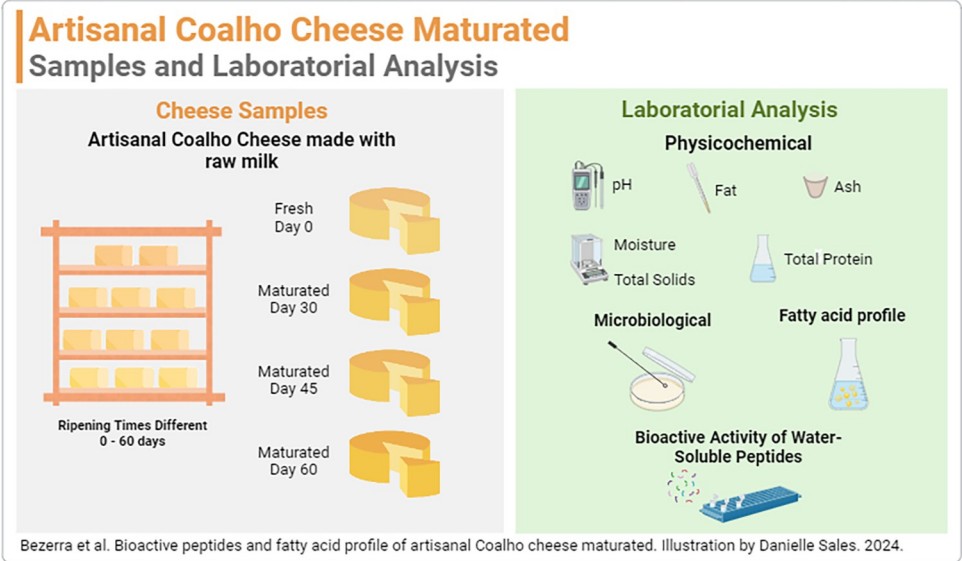

**Fig 2. Metodology illustration.** Schematic diagram of the maturation, samples obtained, and analysis method for Artisanal *Coalho* Cheese.

distilled water. The pH was measured directly after homogenization using a digital potentiometer (Lucadema, São José do Rio Preto, São Paulo), previously calibrated with buffer solutions (pH 4.0 and 7.0), according to the Manual of Physicochemical Analysis of Food of the Adolfo Lutz Institute [29].

## Total solids and moisture

The total solids were determined using a technique described by the Instituto Adolfo Lutz [29]. Adequately sanitized and identified crucibles were placed in an incubator (Lucadema, São José do Rio Preto-SP, Brazil) at 105°C for two hours. After cooling in a desiccator, 3g of cheese sample was weighed, added to the crucibles, and taken to an oven at 105°C for 6 hours. After cooling again in a desiccator, they were reweighed.

The percentage of total solids was determined by the following equation:

$$TS = 100 - (W2 - R)X100/W1$$

Where:
TS = total solids (%)
W1 = initial sample weight (g)
W2 = crucible and sample weights (g)
R = reweight (g).

Moisture (%) was determined by the difference between the initial weight of the sample (W1, g) and the TS (%).

## Fat in total solids

Fat analysis was performed using the Butirometric Method with some adaptations [30]. Three g of macerated cheese samples were added to a Gerber Butyrometer for cheese fat analysis, followed by 9 mL of distilled water at room temperature, 10 mL of sulfuric acid at a density of 1.840 to 1.825 at 20°C, and 1 mL of isoamylic acid. After cleaning the butyrometer edge, the stopper was attached to seal the instrument, and the butyrometers were carefully shaken to dilute the sample. Subsequently, they were centrifuged (centrifuge model 08 BT CENTRIFUGE ITR-KCEN, Porto Alegre) for 10 minutes at 1200x*g*. The results expressed in % of fat were determined according to the level of fat in the butyrometer reading, and the fat in total solids was calculated with the following formula:

$$FTS(\%) = Fx100/TS$$

Where:
FTS = fat in total solids
F = fat (%)
TS = total solids (%).

Non-Fat Solids were obtained through the difference between Total Solids (TS) and Fat (F) [29].

## Protein

Protein content was determined using the Micro-Kjedahl method [31], comprising the digestion, distillation, and titration phases. In the digestion stage, 0.2 g of cheese was weighed in a micro Kjedahl tube, adding 7 mL of prepared digestive solution (cupric sulfate and potassium sulfate) and 5 mL of concentrated sulfuric acid. Then, the sample was digested in a digester block, starting at 50°C and increasing 50°C every 30 minutes until it reached 400°C. After this

time, the solution turned light green, and then the tubes were cooled to begin the distillation step.

For distillation, 8 mL of distilled water was added to the tube, then adapted to the system, neutralizing with approximately 25 mL of 40% NaOH. The distillate was collected in a 250 mL Erlenmeyer containing 10 mL of 4% boric acid (w/v) and indicators. The distillation was completed when 75 mL of the Erlenmeyer was reached.

Subsequently, the titration stage was performed using 0.1 N HCL until the distillate color changed (from green to purple). The spent volume of acid was used to calculate the percentage of total nitrogen in the cheese samples and converted to protein percentage using factor 6.38 (nitrogen conversion factor for milk protein and dairy products).

The percentage of protein was expressed according to the equation:

$$P(\%) = K \ x \ V \ x \ Factor/SM$$

Where:
P = Protein
K = Fc x 0.0014 x 100
Fc = correction factor of sulfuric acid solution
V = volume of hydrochloric acid solution expended on titration
Factor = nitrogen-to-protein conversion factor
SM = sample mass (g)

### Ashes

The ash content of the cheeses was determined according to the methodology of Instituto Adolfo Lutz [29]. Previously cleaned and identified crucibles were placed in an oven (Lucadema, Diadema–SP) for one hour. After cooling in a desiccator, 3 g of sample were weighed and added. Subsequently, they were packed in Muffle Blades (Quimis, Diadema–SP) at 600˚C for 4 hours and reweighed after cooling in a desiccator.

The percentage of ash was determined by the following equation:

$$AC(\%) = R - CW \ x \ 100/SW$$

Where:
AC = ash content (%)
CW = crucible weight (g)
SW = sample weight (g)
R = reweighing (g)

### Fatty acid profile

Samples of 60 g of all cheeses from each maturation period were separated, packed in plastic containers with lids, and then frozen for shipment and determination of the fatty acid profile, according to Kramer et al. [32]. The atherogenicity (AI) and thrombogenicity (TI) indices were calculated according to Ulbricht and Southgate [33]. The cis/trans-18:1 isomers were identified according to their relative elution order under the same chromatographic conditions [34].

### Microbiological analysis

The microbiological analysis was carried out at the Milk Quality Laboratory of the Federal University of Rio Grande do Norte, LABOLEITE/UFRN, according to the methodology

described by Silva [35]. Proportions of each cheese's mass were removed to obtain an aliquot of 25 g of sample, measured on a semi-analytical scale. For cheeses whose rind had already been formed (30, 45, and 60 days of maturation), the portions were obtained after removing about 1 cm of the rind with a sterile knife. Next, the samples were macerated in a previously sterilized porcelain crucible and homogenized with 225 mL of sterile peptone water solution (H2Op) at 0.1%, and 1 mL of this solution was removed for serial dilution in tubes containing 9 mL of sterile peptone water (H2Op) until dilutions of $10^{-1}$ to $10^{-3}$ were obtained. After preparation, 1 mL of the $10^{-1}$ dilution of each sample was seeded in Compact Dry ® plates (Nissui Pharmaceutical Co. Ltda., Tokyo, Japan) specific for analysis of *Salmonella sp.*, *Listeria monocytogenes*, *Staphylococcus aureus*, total and thermotolerant coliforms and incubated according to the manufacturer's guidance. The results evaluated were based on Brasil [36].

## Testing of the bioactive activity of water-soluble peptides

**Extraction of water-soluble peptides.**    The extraction of water-soluble peptides (WSP) was performed according to Lima et al. [37] with some modifications. For extraction, 40 g of each cheese sample was weighed in a 250 mL Erlenmeyer, then homogenized for 10 min with 80 mL of Milli-Q® water (1:2 w/v) at 3000 x*g*. Subsequently, the material was packed in Falcon tubes and homogenized at 3000 x*g* for 5 minutes using a vortex, repeating this procedure three times. The material was then centrifuged at a speed of 7000 rpm for 20 minutes at 4˚C (Hermle Labortechnik Centrifuge 326 HK, Wehingen, Germany). The supernatant containing the water-soluble peptides was collected, and the precipitate was discarded. This process was performed twice to obtain a precipitate-free supernatant. After this procedure, the final supernatant was filtered on quantitative paper (Whatman n. 80) and frozen at -20˚C for subsequent lyophilization. Protein concentrations were determined using bovine serum albumin as standard using the BCA method—Protein Assay Kit (Pierce, Rockford, IL, USA).

**Antioxidant activity assay using 1,1-diphenyl 2-picryl-hydrazil (DPPH•+).**    The antioxidant method using the 1,1-diphenyl-2-picrylhydrazyl radical (DPPH$^{\bullet+}$) was performed following Lima et al. [37]. The concentrations of 5, 2.5, 1.25, 0.625, and 0.312 mg/L of lyophilized samples of water-soluble peptides per mL of distilled water were evaluated.

In a 96-well microplate, 22 μL of each extract at standardized concentrations were mixed with 200 μL of a DPPH$^{\bullet+}$ solution (25 mg/L). Due to the turbidity of the extracts, the whites were prepared after mixing 22 μL of extract with 200 μL of ethanol. The reaction was developed for 2 h in the dark at room temperature, and subsequently, the absorbance was read at 450 and 595 nm using an LM-LGC Microplate Lector (LGC Biotechnologies Ltda., São Paulo, Brazil). Ascorbic acid (vitamin C) was used as a positive control. The concentration of DPPH$^{\bullet+}$ in the reaction medium was calculated from a calibration curve (n = 8; r = 0.991), which was determined using linear regression to deduce further the percentage of DPPH$^{\bullet+}$ remaining (% DPPH$^{\bullet+}$).

The percentage reduction in DPPH$^{\bullet+}$ was calculated according to the equation:

$$AA(\%) = Absi - Abss \; x \; 100/Absi$$

Where:
Absi = Initial absorbance (ethanolic solution + DPPH$^{\bullet+}$)
Abss = Absorbance of the mixture (DPPH$^{\bullet+}$ + mixture)

**Antioxidant assay using 2,2-azine-bis-(3-ethylbenzothiazoline)-6-sulfonic acid radical (ABTS+).**    Antioxidant capacity performed according to the methodology cited by Lima et al. [37], evaluating the concentrations of 5, 2.5, 1.25, 0.625, and 0.312 mg/L of lyophilized samples of water-soluble peptides per mL of distilled water. The oxidation reaction was prepared using

a 7 mM ABTS$^+$ system and stock solution with 140 mM potassium persulfate (final concentration). The mixture was incubated in the dark at room temperature (23 to 25˚C) for 12 to 16 hours. Using a microplate, the ABTS$^+$ solution was diluted in ethanol to an absorbance of 0.7 (± 0.02) at 734 nm in a spectrophotometer. Antioxidant activity was performed using aliquots of 30 μL of samples mixed with 3 mL of diluted ABTS solution. Absorbance at 734 nm was measured in 180 min. Trolox (6-hydroxy-2, 5, 7, 8-tetramethylchrome-2-carboxylic acid) was the reference standard.

The values were calculated and expressed as a percentage of antioxidant activity according to the equation:

$$AA(\%) = Absi - Abss \; x \; 100/Absi$$

Where:
Absi = Initial absorbance (ABTS$^+$ solution)
Abss = Absorbance of the mixture (ABTS$^+$ solution + mixture).

**Determination of angiotensin-converting enzyme inhibitory activity.** The inhibition activity of angiotensin-converting enzyme (ACE) was measured *in vitro* according to the methodology of Cushman et al. [38], with some modifications. Concentrations of 5, 2.5, 1.25, 0.625, and 0.312 mg/L of lyophilized samples of water-soluble peptides per mL of distilled water were used to evaluate this activity. Samples of 20 μL were added to 100 μL of buffered substrate solution at 37˚C; 5 mM hippuryl-L-histidyl-L-leucine (Sigma-Aldrich) in 50 mM HEPES-HCl buffer containing 300 mM NaCl, pH 8.3. The reaction was initiated by adding 40 μL of ACE (0.1 U/mL, rabbit lung, Sigma-Aldrich), incubated at 37˚C for 30 min, and then ended with 150 μL of 1 M HCl. The released hippuric acid was extracted with 1 mL of ethyl acetate, and the organic phase was transferred to a glass tube to be evaporated by heat. The residue was dissolved with 800 μL of distilled water and measured in a spectrophotometer at 228 nm. The antihypertensive Captopril (1 mM, Sigma-Aldrich) was used as a positive control for ACE inhibition.

ACE inhibitory activity was expressed as a percentage using the following formula:

$$ACE - Inhibition(\%) = (A - B)/(A - C) \; x \; 100$$

Where:
ACE-Inhibition = Inhibitory activity (%)
A = Absorbance without Water Soluble Peptides
B = Absorbance without ACE
C = Absorbance in the presence of ACE and soluble peptides.

**Determination of antimicrobial activity.** The antimicrobial activity was carried out according to the methodology of Lima et al. [37], for which strains of *Listeria monocytogenes* ATCC 19117, *Pseudomonas aeruginosa* ATCC 14028, *Escherichia coli* ATCC 25922 and *Salmonella typhimurium* ATCC 14028 were used. All strains were prepared in brain and heart infusion broth (BHI) and aerobically incubated at 37˚C for 18–24 h. Biomass concentration ($1.5 \times 10^8$ CFU/mL) was determined by spectrophotometer measurement of suspension turbidity at 595 nm (Microplate Lector LM-LGC) and then converted into colony-forming units using appropriate calibration curves (turbidity equivalent to 0.5 on the McFarland scale).

All assays to determine antimicrobial activity were performed in a 96-well plate. The controls used were as follows: for the peptides (control of each sample aliquot + liquid culture medium), for the assay (each sample aliquot 50 μL + liquid culture medium 45 μL + each bacterial strain 5 μL), for positive control (each bacterial strain + liquid culture medium) and for negative control (each bacterial strain + liquid culture medium + 10 μg/mL ciprofloxacin).

The microplate was incubated at 37˚C for 18–24 h. Antimicrobial activity was detected by UV-Vis spectrophotometry at an absorbance of 595 nm (LM-LGC Biotechnology, São Paulo, Brazil). The results were expressed as a percentage of antimicrobial activity.

**Experimental design and data analysis.** A completely randomized design was adopted, with 4 maturation periods (0, 30, 45, and 60 days) evaluated in quadruplicate. All data were first tested for homoscedasticity and normality of residuals using Levene and Shapiro-Wilk tests, respectively. The data from the physicochemical analyses and fatty acid profile of the cheeses were tested for linear and quadratic responses using the ordinary least squares regression method by the Jamovi Computer Software [39]. Regarding the bioactivity of the water-soluble peptides, the frequency tables of experimental data (%) are shown as the average of multiple assays. Significance was declared when $P < 0.05$.

## Results and discussion

### Physicochemical properties of cheeses

The results of the physicochemical analyses, including total solids, moisture, solids non-fat (SNF), fat in total solids (FTS), protein, ash, and hydrogen potential (pH) of fresh Artisanal *Coalho* Cheeses (time 0) and those after 30, 45 and 60 days of maturation are presented in Table 2.

A linear increase ($P < 0.05$) in the total solids content and a decrease in the moisture content with the maturation period was observed compared to fresh cheese. Total solids increased from 50.08% at day 0 to 64.41, 68.46, and 69.61% at 30, 45, and 60 days, respectively. During maturation, there is a loss of moisture throughout the process and, consequently, an increase in protein and fat content, i.e., the longer the maturation time, the higher the percentage of total solids due to moisture loss [40]. The fresh cheese in this study was classified as "high moisture", according to current legislation [36]. However, during the maturation process, humidity was reduced, and the cheese was considered to have low moisture content (below 35.9%), differentrom what was established in the technical regulation of identity and quality for Artisanal *Coalho* Cheese [41]. Their results were similar to those of Bonfim [42], who studied the proteolysis of cheese matured by *Enterococcus faecium*. This author observed values of moisture between 52.53 and 54.51% at the initial maturation time, classifying them as "high moisture", while after 60 days of maturation, found a significant reduction in the moisture content with the use of *Enterococcus* and when associated with *Lactococcus lactis*, the cheese presented even lower moisture content, classifying it as "low moisture". Moisture loss during

**Table 2. Analysis of the physicochemical parameters of fresh Artisanal *Coalho* Cheese (day 0) and matured for 30, 45, and 60 days.**

| Characteristics | Time of Maturation (days) | | | | P[1] | |
|---|---|---|---|---|---|---|
| | **0** | **30** | **45** | **60** | **L** | **Q** |
| Total solids (%) | 50.08 ±0.60 | 64.41±1.77 | 68.46±0.78 | 69.61±1.91 | <0.01 | <0.01 |
| Moisture (%) | 49.92±0.60 | 35.59±1.77 | 31.59±0.85 | 30.39±1.91 | <0.01 | <0.01 |
| SNF (%) | 25.94±1.90 | 37.79±1.47 | 40.29±1.04 | 39.91±2.0 | <0.01 | <0.01 |
| FTS (%) | 47.59±2.0 | 41.53±0.45 | 41.16±0.81 | 43.06±1.56 | 0.041 | 0.0002 |
| Protein (%) | 39.23±0.45 | 34.88±1.10 | 33.84±0.95 | 37.78±1.58 | 0.1331 | <0.01* |
| Ash (%) | 6.72±0.35 | 7.38±0.20 | 7.90±0.37 | 7.83±0.55 | 0.0004 | 0.3589 |
| pH value | 5.25±0.13 | 4.88±0.08 | 4.97±0.07 | 4.82±0.02 | 0.0001 | 0.1508 |

SNF: solids-not-fat, FTS: Fat in total solids.

[1]Probability of having a linear (L) or quadratic (Q) effect ($P < 0.05$).

*$y = 0,0047x^2 – 0,3122x + 39,347$ ($R^2 = 0,9229$).

cheese maturation is an intrinsic and common phenomenon related to various process parameters, such as the temperature and relative humidity of the maturation chamber. These factors are paramount for cheeses to acquire peculiar sensory characteristics, resulting in improved flavor, odor, texture, external characteristics, and conservation [43].

In the present study, we observed a linear decrease in FTS content with maturation time ($P < 0.05$). Artisanal *Coalho* Cheese was classified as "fatty" on day 0 (fresh) and "semi-fat" in the following periods of maturation (30, 45, and 60 days) [36]. Despite this variation, the samples from all the maturation periods showed FTS content within the range of the current legislation for Artisanal *Coalho* Cheese, which should be between 35 and 60% [41]. There was an increase in SNF at 30 and 45 days when compared to fresh cheese, with a subsequent reduction at 60 days showing a tendency of a quadratic pattern ($P < 0.05$).

The fat content of cheese is better analyzed when expressed in relation to total solids, preventing variations caused by an eventual loss of moisture. The FTS of this study was lower when compared to the results of Gomes et al. [44], who found an FTS of 52.70% for Artisanal *Coalho* Cheese and 71.57% for industrial *Coalho* Cheese.

There was a quadratic variation in protein content, with a reduction at 30 and 45 days with a subsequent increase at 60 days ($P < 0.01$). The reduction in protein content is due to the maturation process, with the hydrolysis of protein to peptides and amino acids caused by enzyme activity from the lactic acid bacteria present in cheese [45]. The peptides formed from proteolysis have bioactive properties and can influence several beneficial physiological responses in humans [46–49].

Bonfim [42] found a different result when studying the proteolysis of cheese matured by *Enterococcus faecium*. The author found an increase of about 36% to 40% in the protein content after 60 days of maturation, showing a maximum value of 23.71% in one of the treatments. Pereira [50] evaluated the effects of different maturation conditions on the characteristics of artisanal *Minas* Cheese and observed no significant variation in protein content at 7, 14, 21, 42, and 60 days of maturation. The values observed in this study were also higher than those of previous authors studying Artisanal *Coalho* Cheeses. For instance, Silva et al. [23] reported a variation from 20.96 to 22.17%, while Silva et al. [51] found a variation from 26.93 to 29.63%.

The cheese samples from the three maturation periods were more acidic than the fresh samples, with an average pH of 5.25. The values found for the matured cheeses differed from those of other studies on the physicochemical analysis of Artisanal *Coalho* Cheese. Sousa et al. [52] analyzed 104 cheeses collected in several states in Northeast Brazil, reporting an average pH of 5.68 and 5.18 for cheeses with and without health inspection, respectively. The pH of Industrial and Artisanal *Coalho* Cheese ranged from 5.10 to 5.80 in the study by Araújo et al. [53]. Ferreira et al. [21] found values of 5.27 to 5.85 for the pH of Artisanal *Coalho* Cheeses in the State of Pernambuco, also in Brazil.

Reducing cheese pH plays a crucial role in inhibiting the growth of pathogenic bacteria and most microorganisms responsible for cheese spoilage, thus contributing to its safety. The decrease in acidity occurs naturally through the fermentation of milk by lactic acid bacteria, a process influenced by the specific bacterial strains used, besides the production method. Acid production also affects fluid retention and coagulant activity during coagulation; it solubilizes calcium phosphate, thus affecting the texture of the cheese. It promotes syneresis and consequently influences the composition of the cheese and also the activity of enzymes during maturation [54].

Ash content in cheese comprises saline substances and mineral materials recovered from milk, such as calcium, magnesium, zinc, and sodium, and/or added during cheese making [55]. The percentage of ash showed a slight variation among the evaluated maturation periods of 30, 45, and 60 days, with values of 7.38%, 7.90%, and 7.83%, respectively; all higher than that

observed for cheese at time 0. Currently, no Brazilian legislation establishes a minimum value for ash content; however, according to Gomes [56], fresh cheeses have ash percentages between 1.0 and 6.0%, reaching 8.16% in cheese matured for 10 days [57].

## Microbiological analysis of cheeses

The results of the microbiological analyses are presented in Table 3. The Artisanal *Coalho* Cheeses at time 0 presented a microbial load within the maximum requirements established in the current legislation [36]. During the maturation time, 92% of the samples presented satisfactory results for the investigation of microorganisms, which are related to the quality of the raw material used for the production of the cheeses, the use of good manufacturing practices, and the monitoring of the maturation process.

The quality of artisanal cheeses made from raw milk is directly related to milk quality [50]. It is necessary to have healthy cattle, good hygiene practices in milking and milk handling, and efficient hygiene of the equipment and utensils used. In addition, if the cheese is not produced immediately after milking, the milk must be kept refrigerated (from 0 to 4°C) for a maximum of 2 hours from milking to preserve its quality [43].

It is also known that the reduction in the moisture content of matured cheeses also contributes to their better conservation since all microorganisms require high water activity, so most dehydrated cheeses are better preserved even in adverse conditions.

## Fatty acidy profile of cheeses

The fatty acids identified in the samples of fresh Artisanal *Coalho* Cheese (time 0) and matured for 30, 45, and 60 days are presented in Tables 4 and 5.

A total of 27 fatty acids were identified in fresh and matured samples, among which 15 were saturated, 07 monounsaturated, and 05 polyunsaturated. The predominance of saturated

**Table 3. Microbiological analyses of fresh Artisanal *Coalho* Cheese (day 0) and matured for 30, 45, and 60 days.**

| Samples | Coliform/g (35°C)[a] | Coliform/g (45°C)[a] | *Sthaphylococus aureus*/g[a] | *Salmonella* sp./25g | *L. monocytogenes*/ 25g |
|---|---|---|---|---|---|
| FC0A | 2.34 | < 1.00 | < 1.00 | - | - |
| FC0B | 2.00 | < 1.00 | < 1.00 | - | - |
| FC0C | 2.15 | < 1.00 | < 1.00 | - | - |
| FC0D | 2.30 | 2.00 | < 1.00 | - | - |
| MC30A | 2.60 | 1.00 | 3.05 | - | - |
| MC30B | 2.30 | 1.00 | 3.00 | - | - |
| MC30C | 2.30 | 1.00 | 3.00 | - | - |
| MC30D | 2.10 | 1.00 | 2.95 | - | - |
| MC45A | 1.00 | 1.00 | 2.15 | - | - |
| MC45B | 1.00 | 1.00 | 2.84 | - | - |
| MC45C | 1.00 | 1.00 | 2.36 | - | - |
| MC45D | 1.00 | 2.10 | 1.00 | - | - |
| MC60A | < 1.00 | < 1.00 | 1.00 | - | - |
| MC60B | < 1.00 | < 1.00 | < 1.00 | - | - |
| MC60C | < 1.00 | < 1.00 | < 1.00 | - | - |
| MC60D | < 1.00 | < 1.00 | < 1.00 | - | - |

FC0: Fresh cheese (0 days of maturation) considered High Moisture cheese (46 to 55%); MC30: Cheese matured for 30 days; MC45: Cheese matured for 45 days. MC60: Cheese matured for 60 days, all considered Low Moisture cheeses (<36%). -: Absence of microbiological growth.

[a]logCFU/g. The reading value was estimated on plates where the number of colonies was below the minimum limit for the respective microorganism (< 1.00 logCFU/g).

**Table 4. Group of fatty acids present on fresh Artisanal *Coalho* Cheese (day 0) and matured for 30, 45, and 60 days.**

| Group of fatty acids (%) | Time of Maturation (days) | | | | $P^a$ | |
|---|---|---|---|---|---|---|
| | 0 | 30 | 45 | 60 | L | Q |
| SCFA | 6.17±0.12 | 6.12±0.12 | 5.89±0.09 | 5.99±0.38 | 0.3912 | 0.9186 |
| MCFA | 19.32±0.25 | 19.60±0.24 | 19.67±0.08 | 19.92±0.18 | 0.0372 | 0.8206 |
| SFA | 75.29±0.30 | 75.44±0.21 | 74.94±0.40 | 75.68±0.22 | 0.6733 | 0.4586 |
| UFA | 24.71±0.30 | 24.56±0.21 | 25.06±0.40 | 24.32±0.22 | 0.6733 | 0.4586 |
| MFA | 20.40±0.20 | 20.47±0.14 | 20.99±0.34 | 20.39±0.21 | 0.5859 | 0.3511 |
| FMO | 2.81±0.07 | 2.95±0.01 | 2.94±0.01 | 3.03±0.02 | 0.0009 | 0.8213 |
| AI | 3.72±0.07 | 3.86±0.06 | 3.77±0.08 | 3.91±0.06 | 0.1200 | 0.9693 |
| TI | 4.38±0.10 | 4.68±0.12 | 4.58±0.10 | 4.75±0.03 | 0.0081 | 0.5832 |
| n6/n3 | 5.89±0.56 | 8.13±0.51 | 8.91±0.26 | 8.62±0.17 | <0.01 | <0.01 |

SCFA: Short Chain Fatty Acids; MCFA: Medium Chain Fatty Acids; SFA: Saturated Fatty Acids; UFA: Unsaturated Fatty Acids; MFA: Monounsaturated Fatty Acids; FMO: Fatty Acids of Microbial Origin; AI: Atherogenicity Index; TI: Thrombogenicity Index; n6/n3: ratio between n-6 PUFA and n-3 PUFA.
[a]Probability of having a linear (L) or quadratic (Q) effect ($P < 0.05$).

**Table 5. Identified fatty acids on fresh Artisanal *Coalho* Cheese (day 0) and matured for 30, 45, and 60 days.**

| Fatty Acids (%) | Time of Maturation (days) | | | | $P^a$ | |
|---|---|---|---|---|---|---|
| | 0 | 30 | 45 | 60 | L | Q |
| **Saturated** | | | | | | |
| C4:0 | 2.28 ± 0.06 | 2.21± 0.11 | 2.06± 0.06 | 2.06± 0.30 | 0.2866 | 0.9603 |
| C6:0 | 2.33± 0.04 | 2.34± 0.03 | 2.30± 0.04 | 2.33± 0.08 | 0.8419 | 0.9887 |
| C8:0 | 1.56± 0.02 | 1.56± 0.02 | 1.52± 0.02 | 1.59± 0.01 | 0.8137 | 0.2109 |
| C10:0 | 3.46± 0.06 | 3.46± 0.03 | 3.46± 0.03 | 3.61± 0.02 | 0.0658 | 0.0628 |
| C12:0 | 3.92± 0.65 | 3.95± 0.06 | 4.00± 0.02 | 4.07± 0.05 | 0.0611 | 0.4814 |
| C13:0 | 0.10± 0.002 | 0.10± 0.0005 | 0.10± 0.001 | 0.10± 0.0003 | 0.0571 | 0.9429 |
| C13:0 iso | 0.10± 0.0009 | 0.11± 0.002 | 0.11± 0.0003 | 0.19± 0.03 | 0.0189 | 0.0084 |
| C14:0 | 11.93± 0.12 | 12.18± 0.15 | 12.21± 0.03 | 12.25± 0.14 | 0.0556 | 0.5362 |
| C14:0 iso | 0.33± 0.02 | 0.33± 0.006 | 0.32± 0.002 | 0.32± 0.004 | 0.6085 | 0.7801 |
| C14:0 anti | 0.52± 0.05 | 0.48± 0.03 | 0.48± 0.002 | 0.48± 0.004 | 0.2133 | 0.5835 |
| C15:0 | 0.98± 0.13 | 1.13± 0.007 | 1.13± 0.005 | 1.13± 0.097 | 0.0832 | 0.3686 |
| C16:0 | 32.95± 0.06 | 33.54± 0.04 | 33.31± 0.23 | 33.55± 0.14 | 0.0187 | 0.2905 |
| C16:0 anti | 0.20±0.001 | 0.20±0.002 | 0.20±0.004 | 0.20±0.002 | 0.2041 | 0.6631 |
| C17:0 | 0.59±0.001 | 0.61±0.003 | 0.60±0.004 | 0.60±0.005 | 0.0269 | 0.0440 |
| C18:0 | 9.46±0.05 | 9.59±0.14 | 9.46±0.04 | 9.60±0.16 | 0.5083 | 0.9453 |
| **Monounsaturated** | | | | | | |
| C14:1, ω5 | 0.94± 0.01 | 0.94± 0.01 | 0.95± 0.003 | 0.96± 0.01 | 0.122 | 0.5574 |
| C16:1, ω7 | 0.36± 0.004 | 0.37± 0.004 | 0.37±0.003 | 0.36±0.002 | 0.8335 | 0.1071 |
| C16:1 | 1.75±0.007 | 1.78±0.02 | 1.82±0.03 | 1.78±0.01 | 0.1099 | 0.2167 |
| C18:1, t9 | 0.09±0.02 | 0.32±0.02 | 0.32±0.11 | 0.34±0.01 | 0.0001 | 0.0002 |
| C18:1, t11 | 1.15±0.03 | 1.16±0.02 | 1.11±0.01 | 1.08±0.02 | 0.0304 | 0.2291 |
| C18:1, 9 cis | 17.17±0.18 | 17.21±0.13 | 17.67±0.3 | 17.13±0.19 | 0.6767 | 0.3677 |
| C18:1, 11 cis | 0.53±0.10 | 0.54±0.01 | 0.55±0.02 | 0.52±0.01 | 0.8085 | 0.2813 |
| **Polyunsaturated** | | | | | | |
| C18:2, ω6 | 1.68±0.08 | 1.50±0.02 | 1.50±0.04 | 1.40±0.02 | <0.0001 | 0.6963 |
| CLA c9, t11 | 0.33±0.006 | 0.33±0.004 | 0.34±0.08 | 0.34±0.01 | 0.7848 | 0.9318 |
| CLA c12, t10 | 0.03±0.01 | 0.02±0.07 | 0.05±0.01 | 0.04±0.004 | 0.3003 | 0.6667 |
| C18:3, ω3 | 0.35±0.04 | 0.21±0.18 | 0.19±0.004 | 0.18±0.004 | <0.0001 | 0.0325 |
| C20:4 ω6 | 0.31±0.05 | 0.17±0.006 | 0.18±0.01 | 0.18±0.001 | 0.0035 | 0.0457 |

[1]Probability of having a linear (L) or quadratic (Q) effect ($P < 0.05$).

fatty acids was expected since products originating from ruminants naturally contain more of these acids in their composition due to the biohydrogenation process of unsaturated fatty acids in the rumen [58].

The saturated fatty acid with the highest percentage found in both fresh Artisanal *Coalho* Cheese (time 0) and matured cheeses were palmitic acid (C16:0), followed by myristic acid (C14:0). It is known that the former potentially elevates the plasma concentration of cholesterol and LDL-C, and the latter is considered the most effective fatty acid in increasing colisteremia [58].

As for the monounsaturated acids, oleic acid predominated (C18:1 cis-9). This fatty acid produces a beneficial effect by reducing LDL-C levels and inducing lower endogenous cholesterol synthesis compared to polyunsaturated fatty acids [58]. There was no effect of maturation time on the content of these most predominant fatty acids in Artisanal *Coalho* Cheese.

These results are consistent with Beltrao et al. [59], who analyzed the fatty acid profile of bovine, goat, and mixed milk used to manufacture symbiotic chevrotim cheese. They found that the fatty acids C:14, C:16, C:18, and C:18:1, n-9 trans had a high percentage, being the same ones that presented a higher percentage in the matured cheeses of this study.

The human body cannot synthesize omega-6 and omega-3 fatty acids, so they are essential and should be part of the diet [60]. However, the balance in the intake of both is important because the high consumption of omega-6 associated with the low consumption of omega-3 may be related to the development of several cardiovascular pathologies [60, 61]. It was observed that maturation provided a linear reduction in C18:2 n-6, C18:3 n-3, and C20:4 n-6 concentrations. The hydrogenation of unsaturated fatty acids by bacteria is already known in the rumen environment [62]. It could also explain the decrease in the percentage of polyunsaturated fatty acids as the cheese matures with a concomitant linear increase in the content of fatty acids from microbial origin (FMO) presented in Table 4.

There was no effect of maturation time on CLA contents. CLA is known to have several health benefits, such as helping to promote weight loss, preventing cancer and atherosclerosis, and having immune properties. A study in the United States indicates that milk has a CLA content of 0.63% to 1.16% and in cheeses of 0.40 to 1.70% of the total fatty acids [63]. In the study by Gonçalves [64], there was no significant change in the CLA levels of Minas Padrão Cheese when submitted to 20 and 60 days of maturation. Although the literature mentions positive effects related to its consumption, studies are being conducted in search of the necessary amount of daily intake for CLA to exert its beneficial effects [65].

The Atherogenicity Index (AI) indicates the relationship between the primary pro-atherogenic saturated fatty acids (C12:0, C14:0, and C16:0) and the unsaturated fatty acids considered anti-atherogenic because they inhibit plaque accumulation and reduce the levels of phospholipids, cholesterol, and esterified fatty acids. Therefore, ingesting foods with lower FI values may benefit human health [10].

The Thrombogenicity Index (TI) evaluates the potential of fatty acids to form clots in blood vessels by the ratio between pro-thrombogenic fatty acids (C12:0, C14:0, and C16:0), antithrombogens, monounsaturated fatty acids and omega 3 and 6. Therefore, the lower the TI of the food, the greater the benefit to consumers' health [10]. In this study, there was an increase in TI with the increase in the number of days of the maturation period evaluated. Fresh Artisanal *Coalho* Cheese presented an IT of 4.38%, changing to 4.68% at 30 days, 4.58% at 45 days, and 4.75% at 60 days of maturation. Chen et al. [10] demonstrated that bovine milk presented values ranging from 2.05 to 4.65, and Salles et al. [66] showed a variation from 4.51 to 4.65 for milk from Jersey animals. Both the IA and the TI of the cheeses in the experiment were considered unsatisfactory since values below 1.0 are desirable [33].

The n-6/n-3 ratio is associated with fatty acids essential to human nutrition, composed of linoleic acid and α-linolenic acid, among others. The appropriate ratio for obtaining health benefits should be around 2–5:1, and excessive amounts of n-6 can increase the risk of cancer, cardiovascular diseases, and inflammatory and autoimmune diseases [67]. The maturation time provided the cheeses with a linear and quadratic effect on the n-6/n-3 ratio, and in all periods, the proportion was higher than recommended by the literature.

Milk fat contains a small amount of odd- and branched-chain fatty acids originating from metabolites synthesized by rumen bacteria, including mainly saturated fatty acids with one or more methyl branches at the iso or anteiso position [11]. Bacteria are also composed of these fatty acids, making it possible for their identification to be considered a bacterial biomarker [68]. The present study reported a linear effect on FMO, which may represent bacterial multiplication over the maturation period.

## Antioxidant activity of peptides

It was found that all peptide extracts showed antioxidant activity at concentrations of 5, 2.5, 1.25, 0.625, and 0.312 mg/L. However, the peptide concentration of 5 mg/L showed a greater scavenging capacity against DPPH$^{\bullet+}$ and ABTS$^+$ radicals, as shown in Table 6.

Extracts with 5 mg/L showed an increase in antioxidant capacity with maturation, with greater activity in the samples of cheeses matured for 45 days, with 82.69% in the DPPH$^{\bullet+}$ methodology and 72.66% in the ABTS$^+$. The difference found between the scavenging capacity of DPPH$^{\bullet+}$ and ABTS$^+$ radicals may be explained by the reaction between the radical and the peptides that may occur differently due to the stability between the radical and the peptides [69]. This result even surpasses the antioxidant capacity of several citrus fruits recognized for being rich in antioxidant substances such as ascorbic acid, phenolic compounds, flavonoids, and limonoids that are important for human nutrition [70]. Couto et al. [71] found an antioxidant capacity, through the DPPH$^{\bullet+}$ methodology, of 66.24% of lime orange juice, 60.32% of Bahia orange, 49.15% of pear orange, and only 12.78% of murcote mandarin.

It was also observed that there was a decline of 12.36% and 18.62% in antioxidant capacity when the Artisanal *Coalho* Cheese reached 60 days of maturation. This leads us to believe that

**Table 6. Percentage of antioxidant activity (DPPH$^{\bullet+}$ and ABTS$^+$) of peptides extracted from fresh Artisanal *Coalho* Cheese (day 0) and matured for 30, 45, and 60 days.**

| Antioxidant activity (%)[a] | Time of Maturation (days) | | | |
|---|---|---|---|---|
| | 0 | 30 | 45 | 60 |
| **DPPH$^{\bullet+}$** | | | | |
| 5 | 65.29 | 76.8 | 82.69 | 72.47 |
| 2.5 | 58.01 | 72.19 | 75.63 | 65.84 |
| 1.25 | 51.01 | 67.31 | 68.94 | 61.57 |
| 0.625 | 44.84 | 61.76 | 57.73 | 55.19 |
| 0.312 | 37.51 | 57.43 | 50 | 44.29 |
| **ABTS$^+$** | | | | |
| 5 | 57.43 | 78.62 | 72.86 | 59.29 |
| 2.5 | 38.95 | 69.67 | 68.29 | 52.1 |
| 1.25 | 28.62 | 56.19 | 47.43 | 46.38 |
| 0.625 | 24.24 | 45.95 | 37.38 | 33.29 |
| 0.312 | 16.62 | 36.24 | 26.14 | 23.29 |

[a]Activities carried out in concentrations of 5, 2.5, 1.25, 0.625, and 0.312 mg/L of water-soluble peptides.

the concentration of bioactive peptides increases with the maturation of cheeses but decreases when proteolysis reaches a certain level, thus reducing the concentration of peptides for this bioactivity [72].

Based on these results, we found that matured Artisanal *Coalho* Cheese presents an important benefit to consumers and industry through its antioxidant potential since the increase in oxidative stress is involved in most chronic diseases and aging of living organisms, in addition to being one of the causes of the decrease in the shelf life of food and raw materials in general [73, 74].

The result was similar to Silva et al. [51], who reported that Artisanal *Coalho* Cheese from different regions of the state of Pernambuco showed antioxidant activity between 91.1% and 75.92% for ABTS$^+$ and DPPH$^{•+}$, respectively. Minas frescal cheese also showed radical sequestration capacity, with a sequestration of 86.13% in the ABTS$^+$ methodology and 85.83% in the DPPH$^{•+}$ [75].

The antioxidant activity of bioactive peptides was also observed in Cheddar Cheese [76, 77], Fresco [78], Parmigiano-Reggiano [79], cottage cheese [80], cheeses made with raw sheep's milk and matured in different periods (Feta, Pecorino Toscano, Roquefort, Pecorino Sardo and Cerrilhano) [81] and Mexican white cheese [82].

## ACE inhibitory activity of peptides

Table 7 shows the percentage of ACE inhibition capacity of water-soluble peptide extracts obtained from fresh Artisanal *Coalho* Cheese (time 0) and matured for 30, 45, and 60 days. The concentration of the extracts at 5 mg/L was the one that best demonstrated ACE inhibitory capacity, with higher ACE inhibitory activity in cheese matured for 45 days, with subsequent reduction.

The correlation between maturation and the presence of antihypertensive peptides was also demonstrated by Meisel et al. [72], who reported maximum inhibition of ACE in Gouda cheese matured for three months, compared to the same cheese whose maturation was less than three months or for a more extended period.

ACE inhibitor peptides have been isolated from Italian cheeses such as Crescenza and Gorgonzola [83]. This inhibitory activity was also evidenced in other varieties of cheese such as Gouda, Blue, Edam, and Havarti, which, in a feeding experiment in hypertensive rats (SHR), resulted in a decrease in systolic blood pressure 6 hours after gastric intubation [84]. In a study carried out by Iwaniak et al. [85], it was shown that Gouda, even though it was made with different levels of β-casein, had an inhibitory action on ACE, being considered a predominant bioactivity in this cheese, occurring only due to the presence of peptides in its composition and not linked to its quantity.

**Table 7. Percentage of ACE inhibitory activity of the extracted peptides from fresh Artisanal *Coalho* Cheese (day 0) and matured for 30, 45, and 60 days.**

| Concentrations (mg/L)[a] | Time of Maturation (days) | | | |
|---|---|---|---|---|
| | **0** | **30** | **45** | **60** |
| 5 | 43.96 | 59.49 | 67.38 | 61.06 |
| 2.5 | 38.81 | 57.58 | 64.3 | 57.64 |
| 1.25 | 30.57 | 54.4 | 61.77 | 55.36 |
| 0.625 | 27.43 | 52.8 | 59.16 | 52.32 |
| 0.312 | 23.69 | 46.59 | 53.74 | 49.99 |

[a]Activities carried out in concentrations of 5, 2.5, 1.25, 0.625, and 0.312 mg/L of water-soluble peptides.

In Manchego cheese produced from sheep milk, the ACE inhibitory activity showed a different evolution throughout the maturation period, decreasing in the first four months, with a subsequent increase and subsequent decrease in cheese at 12 months [86].

Peptides with antihypertensive action were also detected in cheeses made with raw sheep's milk produced in southern Brazil, such as the Roquefort Type, the Feta Type, and the Pecorino Toscano Type cheeses matured for 60, 180, and 270 days. As well as in Uruguayan cheeses, such as Pecorino Sardo, with 80, 120, and 160 days of maturation, and Cerrilano matured for 90 and 120 days [81].

The antihypertensive peptides isoleucine-proline-proline (Ile-Pro-Pro) and valine-proline-proline (Val-Pro-Pro) were found in Swiss cheeses (Appenzeller, Tilsiter, Tête de Moine, Vacherin fribourgeois, Emmental, Gruyère and Berner Hobelkäse) and increased during the maturation process, reaching 100 mg/kg after 4 to 7 months [87].

Although several studies have demonstrated the functionality of peptides in the cardiovascular system, most experiments have analyzed the *in vitro* capacity rather than the signaling mechanisms of action. Therefore, more multidisciplinary research and knowledge (biology, medicine, bioinformatics, and food technology) is needed [8].

### Antimicrobial activity of peptides

Table 8 presents the results observed for the water-soluble peptide extracts regarding their antimicrobial activity in the different maturation periods of Artisanal *Coalho* Cheese. The extract obtained from fresh cheese did not show antimicrobial activity for *Escherichia coli*, and the extract from cheese matured for 30 days at the two lowest concentrations tested. For all

**Table 8. Percentage of antimicrobial activity of extracted peptides from fresh Artisanal *Coalho* Cheese (day 0) and matured for 30, 45, and 60 days.**

| Microorganism | WSP[a] | Time of Maturation (days) | | | |
|---|---|---|---|---|---|
| | | **0** | **30** | **45** | **60** |
| *Listeria monocytogenes* | 5 | 31.16 | 53.20 | 78.62 | 89.61 |
| | 2.5 | 21.74 | 43.91 | 69.76 | 77.17 |
| | 1.25 | 14.90 | 36.85 | 54.95 | 65.69 |
| | 0.625 | 9.51 | 30.94 | 50.39 | 60.87 |
| | 0.312 | 4.91 | 22.74 | 41.85 | 49.47 |
| *Pseudomonas aeruginosa* | 5 | 37.691 | 81.564 | 88.305 | 98.18 |
| | 2.5 | 24.320 | 50.652 | 60.447 | 73.78 |
| | 1.25 | 13.780 | 45.065 | 48.454 | 67.67 |
| | 0.625 | 4.507 | 34.264 | 40.000 | 56.57 |
| | 0.312 | 1.713 | 21.862 | 32.849 | 37.06 |
| *Escherichia coli* | 5 | - | 43.02 | 77.00 | 91.72 |
| | 2.5 | - | 26.23 | 68.08 | 65.06 |
| | 1.25 | - | 11.42 | 57.09 | 57.99 |
| | 0.625 | - | - | 46.20 | 27.08 |
| | 0.312 | - | - | 40.52 | 15.61 |
| *Salmonela typhimurium* | 5 | 36.79 | 71.50 | 77.64 | 92.58 |
| | 2.5 | 30.76 | 48.09 | 70.53 | 85.92 |
| | 1.25 | 22.40 | 38.87 | 63.21 | 80.76 |
| | 0.625 | 15.22 | 30.34 | 56.24 | 72.99 |
| | 0.312 | 8.84 | 26.14 | 49.10 | 66.64 |

[a]WSP: Water-Soluble Peptides. Activities carried out in concentrations of 5, 2.5, 1.25, 0.625, and 0.312 mg/L.

-: Absence of antimicrobial activity.

other extracts, antimicrobial activity was observed against all strains studied, and this activity increased as the maturation time increased. This result confirms that bactericidal and/or bacteriostatic activity peptides are released during cheeses' maturation. Extracts of 5 mg/L of cheeses matured for 60 days achieved antimicrobial activity higher than 89% for *Listeria monocytogenes*, *Escherichia coli*, and *Salmonella typhimurium*, reducing the growth of *Pseudomonas aeruginosa* by 98%.

Other cheeses also showed antimicrobial potential, such as Artisanal *Coalho* Cheese from Pernambuco, in which significant antimicrobial activity inhibited *Bacillus subtilis* and *Enterococcus faecalis* [51]. Canastra cheese matured at 09, 23, and 30 days showed peptide fractions with antimicrobial activity against *Escherichia coli* ATTCC25922 [88]. The peptides of minas frescal cheese were able to inhibit the growth of *Bacillus subtilis* and *Enterococcus faecalis* in all concentrations tested, ranging from 0.78 to 25 mg/mL, and in the tests with *Staphylococcus aureus*, *Escherichia coli* and *Pseudomonas aeruginosa* it was observed minimum inhibitory concentration of 6.25 mg/mL, 3.13 mg/mL and 3.13 mg/mL, respectively [75].

Antimicrobial peptides were also isolated in Mozzarella, Itálico, Crescenza, and Gorgonzola cheeses, with a specific inhibitory action for the endopeptidases of *Pseudomonas fluorescens*, a microorganism responsible for compromising the technological and organoleptic characteristics of dairy products [83]. Peptides with antimicrobial *in vitro* bioactivity have also been identified in Emmental cheese [89].

Lima et al. [90] researched bioactive peptides in Artisanal *Coalho* Cheese and found, through mass spectrometry analysis, the presence of peptides that the literature already confirmed to inhibit Gram-negative and positive bacteria. The presence of fragments similar to isracidin, a peptide derived from αs1-CN treated with chymosin, and the N-terminal fragment of this protein (f1-23) that acts against Gram-positive bacteria was detected [91]. The presence of αs1-casein residue (f 99–109) was also confirmed, which has antimicrobial activity of Gram-positive bacteria such as *B. subtilis* and Gram-negative bacteria such as *E. coli* [92]. The fragments (f181-207), (f175-207), and (f164-207) have also been observed to have growth inhibition of Gram-positive and negative microorganisms [91].

In general, these peptides act in association with the lipids of the plasma membrane of microorganisms, promoting increased permeability. Because they have a positive electrical charge and are amphipathic, they have solubility in an aqueous environment, thus being easily inserted into lipid membranes and triggering the death of the target microorganisms [93].

The peak of the antimicrobial potential at 60 days of maturation may also explain the decrease in the antioxidant and inhibitory potential of ACE in the same period since water-soluble peptides have effects against Gram-positive and Gram-negative bacteria, being able to inactivate bacteria of dairy origin, necessary in the breakdown of proteins and release of peptides with antioxidant and antihypertensive action.

Bioactive peptides with antimicrobial activity may be an alternative to conventional antibiotic therapy due to the advent of microorganisms resistant to antibiotics currently used in animal production. Their presence can be considered a protective factor during the production process since it enables a possible reduction of contamination, and together with good production practices, it can contribute to increasing the shelf life of dairy products. Therefore, they become interested in the product as agents to control microbial contamination and maintain consumer health [90, 94]. Fig 3 shows the summary of the principal results obtained in the study.

## Conclusions

Wooden shelf maturation (up to 60 days) of Coalho cheese improves the concentration of bioactive peptides without changing the percentage of total saturated fatty acids. However,

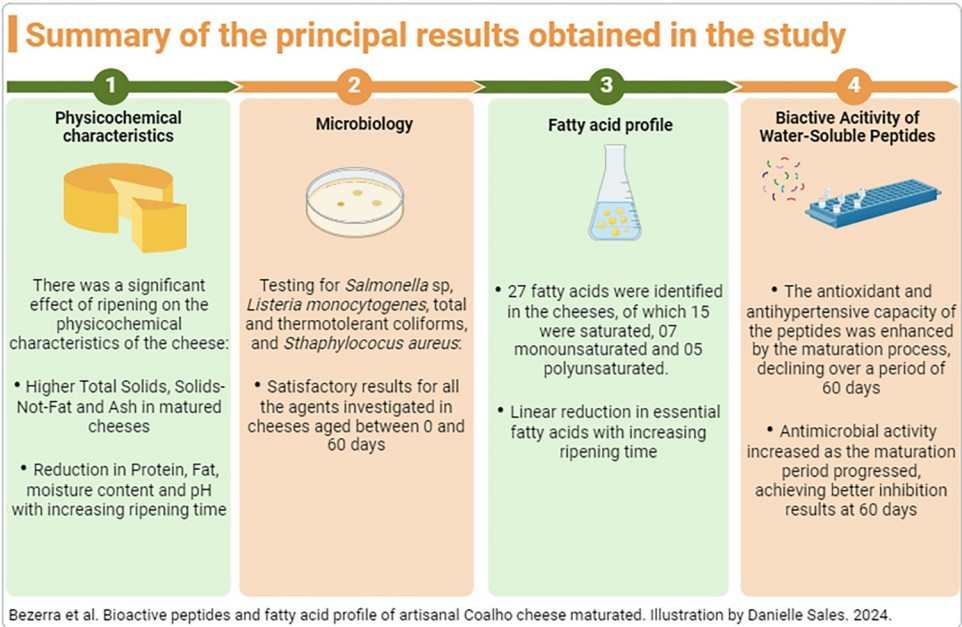

**Fig 3. Results illustration.** Summary of the principal results obtained in the study.

polyunsaturated fatty acids, important in preventing cardiovascular diseases, decrease with maturation without significant changes in CLA concentration. The peptide fractions obtained from Artisanal *Coalho* Cheese showed antihypertensive bioactive potential with ACE inhibition, antioxidant, and antimicrobial, which are beneficial properties to consumers' health that are enhanced by the maturation process. Also, antimicrobial activity is a protective factor of cheese, and it increases shelf life since it makes it possible to reduce contamination in the product along with good production practices.

## Supporting information

**S1 File. Supplementary tables and figures.** Description of the study variables and observed data: Physicochemical parameters, Fatty acid profile, Activity of water-soluble peptides extracted from fresh Artisanal Coalho Cheese (Time 0) and matured for 30, 45 and 60 days. (DOCX)

## Acknowledgments

The authors would like to thank the D.C.S. and E.d.O.M.A.'s agency, Conselho Nacional de Desenvolvimento Científico e Tecnológico (CNPq), and Fundação de Amparo e Promoção da Ciência, Tecnologia e Inovação do Rio Grande do Norte (FAPERN). Figs 1–3 were created in Biorender.com (accessed on October 24, 2023).

## Author Contributions

**Conceptualization:** Ana Lúcia F. Porto, Adriano H. N. Rangel.

**Data curation:** Débora A. F. V. A. Bezerra, Cláudio V. D. M. Ribeiro.

**Formal analysis:** Débora A. F. V. A. Bezerra, Cláudio V. D. M. Ribeiro.

**Investigation:** Débora A. F. V. A. Bezerra, Karoline M. S. Souza, Danielle C. Sales, Cláudio V. D. M. Ribeiro.

**Methodology:** Débora A. F. V. A. Bezerra, Karoline M. S. Souza, Cláudio V. D. M. Ribeiro, Ana Lúcia F. Porto, Adriano H. N. Rangel.

**Project administration:** Ana Lúcia F. Porto, Adriano H. N. Rangel.

**Supervision:** Ana Lúcia F. Porto, Adriano H. N. Rangel.

**Writing – original draft:** Débora A. F. V. A. Bezerra, Karoline M. S. Souza.

**Writing – review & editing:** Danielle C. Sales, Emmanuella O. M. Araújo, Stela A. Urbano, Claudio Cipolat-Gotet, Katya Anaya, Cláudio V. D. M. Ribeiro, Ana Lúcia F. Porto, Adriano H. N. Rangel.

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
