## [Decision Letter · Decision Letter 0]

22 Apr 2024

PONE-D-24-12456Effect of ripening time on content of bioactive peptides and fatty acids profile of Artisanal Coalho cheesePLOS ONE

Dear Dr. Sales,

Thank you for submitting your manuscript to PLOS ONE. After careful consideration, we feel that it has merit but does not fully meet PLOS ONE’s publication criteria as it currently stands. Therefore, we invite you to submit a revised version of the manuscript that addresses the points raised during the review process. Please submit your revised manuscript by Jun 06 2024 11:59PM. If you will need more time than this to complete your revisions, please reply to this message or contact the journal office at plosone@plos.org. Please include the following items when submitting your revised manuscript:A rebuttal letter that responds to each point raised by the academic editor and reviewer(s). You should upload this letter as a separate file labeled 'Response to Reviewers'.A marked-up copy of your manuscript that highlights changes made to the original version. You should upload this as a separate file labeled 'Revised Manuscript with Track Changes'.An unmarked version of your revised paper without tracked changes. You should upload this as a separate file labeled 'Manuscript'.

We look forward to receiving your revised manuscript.

Kind regards,

Guadalupe Virginia Nevárez-Moorillón, Ph.D.

Academic Editor

PLOS ONE

Journal Requirements:

**Additional Editor Comments:**

Please carefully revise the suggestions done by both reviewers.

Reviewers' comments:

Reviewer's Responses to Questions

**Comments to the Author**

1. Is the manuscript technically sound, and do the data support the conclusions?

Reviewer #1: Yes

Reviewer #2: Partly

2. Has the statistical analysis been performed appropriately and rigorously? 

Reviewer #1: Yes

Reviewer #2: No

3. Have the authors made all data underlying the findings in their manuscript fully available?

Reviewer #1: Yes

Reviewer #2: Yes

4. Is the manuscript presented in an intelligible fashion and written in standard English?

Reviewer #1: No

Reviewer #2: Yes

5. Review Comments to the Author

Reviewer #1: After reviewing the submitted manuscript titled “Effect of ripening time on content of bioactive peptides and fatty acids profile of Artisanal Coalho cheese” I have the following observations and comments:

1. Line 33 “with 04 maturation periods” I suggest replacing number with word four.

2. I would recommend replacing the keywords with more appropriate ones.

3. In lines 99-100, could you specify more where the farm is located.

4. Lines 117-119 what was the temperature of the milk used?

5. Figure 1 lack technological parameters like time, temperature for certain technological processes.

6. Why during microbiological analysis LAB counts, enterobacteria and total aerobic bacteria counts were not determined?

7. Line 430 why do you state that the results are in CFU/mL and not CFU/g?

8. What est. in table 3 means and why do you provide it?

9. Please provide the results in table 3 in log CFU

10. Please improve the reference list by carefully providing all the doi information and writing the names in Italic where it is necessary.

11. English language should be improved in some parts of the text.

When these points will be adjusted, I would recommend this manuscript for publication.

Reviewer #2: Remarks:

The aim of this manuscript is very interesting for the readers; however, I think that this manuscript must be improved, mainly the Materials and Methods and Conclusions sections.

lines 70-72: please modify this sentence according to new scientific results!!

Table 1: how many samples were investigated? Are these parameters originated directly from each cheese making batch?

Fig 1: the cheese making process is very draft, please give more information about cheese making! Maybe create a Table!

line 130: Hoe many batches were used during the cheese making? Or 16 samples originated from a batch?

line 140: Physio-chemical analysis or Physicochemical analysis?

line 221: please add the reference about AI and TI!

line 328: please improve this subsection! How investigated the data distribution and homogeneity? What method was used: ANOVA or linear model? Did used linear and quadratic effect? I think used not only linear regression…!

Table 4: FMO: it means odd fatty acids?

Conclusions: please rewrite this section: the lines 645-649 are not good for conclusion, line 649 is too general phrase! Acceptable lines are only 650-656!

6. PLOS authors have the option to publish the peer review history of their article (what does this mean?). If published, this will include your full peer review and any attached files.

Reviewer #1: No

Reviewer #2: **Yes: **Ferenc Pajor

---

## [Author Response · Author response to Decision Letter 0]

11 Jun 2024

June 11, 2024.

Plos One 

Dear Reviewers

Thank you very much for taking the time to review this manuscript “Effect of ripening time on the content of bioactive peptides and fatty acids profile of Artisanal Coalho cheese”.

Their comments were fully appreciated. The manuscript was revised, and a broader text reformulation was carried out. We also did another round of editing in professional languages. All edits suggested, and other revisions are marked in the Manuscript.

Review Comments to the Author - Reviewer #1:

Reviewer #1: After reviewing the submitted manuscript titled “Effect of ripening time on content of bioactive peptides and fatty acids profile of Artisanal Coalho cheese” I have the following observations and comments:

1. Line 33 “with 04 maturation periods” I suggest replacing number with word four.

Response: Changed as suggested.

2. I would recommend replacing the keywords with more appropriate ones.

Response: Keywords were changed with words that are not in the title.

3. In lines 99-100, could you specify more where the farm is located.

Response: Information added.

4. Lines 117-119 what was the temperature of the milk used?

Response: 35°C. Added information.

5. Figure 1 lack technological parameters like time, temperature for certain technological processes.

Response: Corrected figure with information.

6. Why during microbiological analysis LAB counts, enterobacteria and total aerobic bacteria counts were not determined?

Response: The microbiological analysis was used to understand the safety of the cheese after processing. We respected the official microbiological requirements in Brazil, which consider the level of contamination by Salmonella sp., Listeria monocytogenes, Staphylococcus aureus, total (35 ºC) and thermotolerant (45 °C) coliforms (Brasil. Ordinance No. 146 of March 7, 1996. Technical Regulation on the Identity and Quality of Dairy Products. Ministry of Agriculture and Livestock. Official Gazette of the Federative Republic of Brazil, Brasília. 1996). 

7. Line 430 why do you state that the results are in CFU/mL and not CFU/g?

Response: The correct CFU/g. There was a typo in the unit. Thank you for identifying it.

8. What est. in table 3 means and why do you provide it?

Response: “est.” means “estimated”. We use it to report that the direct count on the plates was below the lower limit of colonies for that microorganism. We understand that the expression “est.” can confuse the reader. We have, therefore, removed it. 

9. Please provide the results in table 3 in log CFU

Response: Thank you for the suggestion. We have changed this in the table.

10. Please improve the reference list by carefully providing all the doi information and writing the names in Italic where it is necessary.

Response: The reference list has been revised as suggested. 

11. English language should be improved in some parts of the text.

Response: The English language has been revised. 

When these points will be adjusted, I would recommend this manuscript for publication.

Review Comments to the Author - Reviewer #2:

Reviewer #2: Remarks:

The aim of this manuscript is very interesting for the readers; however, I think that this manuscript must be improved, mainly the Materials and Methods and Conclusions sections.

lines 70-72: please modify this sentence according to new scientific results!!

Response: Thank you for the suggestion. We have changed this.

Table 1: how many samples were investigated? Are these parameters originated directly from each cheese making batch?

Response: These parameters were obtained from just one batch of milk intended for cheese production. We have supplemented the information in the text.

Fig 1: the cheese making process is very draft, please give more information about cheese making! Maybe create a Table!

Response: Figure 1 has been supplemented with manufacturing information, mainly the time and temperature of the stages.

line 130: Hoe many batches were used during the cheese making? Or 16 samples originated from a batch?

Response: Sixteen pieces of cheese were produced from one batch of milk.

line 140: Physio-chemical analysis or Physicochemical analysis?

Response: The correct one is Physicochemical analysis.

line 221: please add the reference about AI and TI!

Response: Information added. Thank you for the suggestion.

line 328: please improve this subsection! How investigated the data distribution and homogeneity? What method was used: ANOVA or linear model? Did used linear and quadratic effect? I think used not only linear regression…!

Response: The term linear regression refers to linear models in which the parameters to be estimated are additive (as opposed to non-linear models). Therefore, any polynomial regression, such as linear and quadratic regressions, may be included. We understand that the text was not clear, and it was changed as suggested. 

We also tested the assumptions to analyze regression models. They were also included in the text.

Table 4: FMO: it means odd fatty acids?

Response: FMO was defined in the footnote of Table 4. However, this definition is located on the next page (lines 440-441).

Conclusions: please rewrite this section: the lines 645-649 are not good for conclusion, line 649 is too general phrase! Acceptable lines are only 650-656!

Response: Reviewed. Thank you for the suggestion.

Best regards, 

Danielle C. Sales, PhD

Corresponding author

Academic Unit Specialized in Agricultural

Federal University of Rio Grande do Norte (UFRN)

Macaiba, Rio Grande do Norte, Brazil

daniellecsales@hotmail.com

---

## [Editor Report · Decision Letter 1]

20 Jun 2024

Effect of ripening time on the content of bioactive peptides and fatty acids profile of Artisanal Coalho cheese

PONE-D-24-12456R1

Dear Dr. Sales,

We’re pleased to inform you that your manuscript has been judged scientifically suitable for publication and will be formally accepted for publication once it meets all outstanding technical requirements.

Kind regards,

Guadalupe Virginia Nevárez-Moorillón, Ph.D.

Academic Editor

PLOS ONE
---

## [Editor Report · Acceptance letter]

27 Jun 2024

PONE-D-24-12456R1 

PLOS ONE

Dear Dr. Sales, 

I'm pleased to inform you that your manuscript has been deemed suitable for publication in PLOS ONE. Congratulations! Your manuscript is now being handed over to our production team.

Kind regards, 

on behalf of

Dr. Guadalupe Virginia Nevárez-Moorillón 

Academic Editor

PLOS ONE